# Numerical influence of ReLU'(0) on backpropagation

**David Bertoin**
IRT Saint Exupéry
ISAE-SUPAERO
ANITI
Toulouse, France
david.bertoin@irt-saintexupery.com

**Jérôme Bolte**
Toulouse School of Economics
Université Toulouse 1 Capitole
ANITI
Toulouse, France
jbolte@ut-capitole.fr

**Sébastien Gerchinovitz**
IRT Saint Exupéry
Institut de Mathématiques de Toulouse
ANITI
Toulouse, France
sebastien.gerchinovitz@irt-saintexupery.com

**Edouard Pauwels**
CNRS
IRIT, Université Paul Sabatier
ANITI
Toulouse, France
edouard.pauwels@irit.fr

## Abstract

In theory, the choice of $\text{ReLU}'(0)$ in $[0, 1]$ for a neural network has a negligible influence both on backpropagation and training. Yet, in the real world, 32 bits default precision combined with the size of deep learning problems makes it a hyperparameter of training methods. We investigate the importance of the value of $\text{ReLU}'(0)$ for several precision levels (16, 32, 64 bits), on various networks (fully connected, VGG, ResNet) and datasets (MNIST, CIFAR10, SVHN, ImageNet). We observe considerable variations of backpropagation outputs which occur around half of the time in 32 bits precision. The effect disappears with double precision, while it is systematic at 16 bits. For vanilla SGD training, the choice $\text{ReLU}'(0) = 0$ seems to be the most efficient. For our experiments on ImageNet the gain in test accuracy over $\text{ReLU}'(0) = 1$ was more than 10 points (two runs). We also evidence that reconditioning approaches as batch-norm or ADAM tend to buffer the influence of $\text{ReLU}'(0)$'s value. Overall, the message we convey is that algorithmic differentiation of nonsmooth problems potentially hides parameters that could be tuned advantageously.

## 1 Introduction

**Nonsmooth algorithmic differentiation:** The training phase of neural networks relies on first-order methods such as Stochastic Gradient Descent (SGD) [14, 9] and crucially on algorithmic differentiation [15]. The fast "differentiator" used in practice to compute mini-batch descent directions is the backpropagation algorithm [29, 4]. Although designed initially for differentiable problems, it is applied indifferently to smooth or nonsmooth networks. In the nonsmooth case this requires surrogate derivatives at the non regularity points. We focus on the famous $\text{ReLU} := \max(0, \cdot)$, for which a value for $s = \text{ReLU}'(0)$ has to be chosen. A priori, any value in $[0, 1]$ bears a variational sense as it corresponds to a subgradient [27]. Yet in most libraries $s = 0$ is chosen; it is the case for TensorFlow [2], PyTorch [26] or Jax [10]. Why this choice? What would be the impact of a different value of $s$? How this interacts with other training strategies? We will use the notation $\text{backprop}_s$ to denote backpropagation implemented with $\text{ReLU}'(0) = s$ for any given real number $s$.[1]

---

[1]Definition in Section 2. This can be coded explicitly or cheaply emulated by $\text{backprop}_0$. Indeed, considering $f_s \colon x \mapsto \text{ReLU}(x) + s(\text{ReLU}(-x) - \text{ReLU}(x) + x) = \text{ReLU}(x)$, we have $\text{backprop}_0[f_s](0) = s$.

35th Conference on Neural Information Processing Systems (NeurIPS 2021).

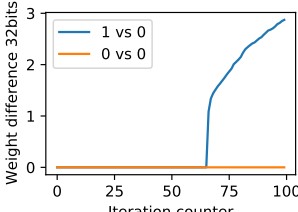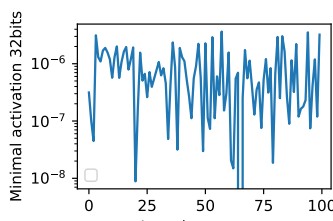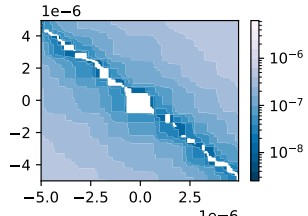

Figure 1: Left: Difference between network parameters ($L^1$ norm), 100 iterations within an epoch. "0 vs 0" indicates $\|\theta_{k,0} - \tilde{\theta}_{k,0}\|_1$ where $\tilde{\theta}_{k,0}$ is a second run for sanity check, "0 vs 1" indicates $\|\theta_{k,0} - \theta_{k,1}\|_1$. Center: minimal absolute activation of the hidden layers within the $k$-th mini-batch, before ReLU. At iteration 65, the jump on the left coincides with the drop in the center, and an exact evaluation of $\text{ReLU}'(0)$. Right: illustration of the bifurcation zone at iteration $k = 65$ in a randomly generated network weight parameter plane (iteration 65 in the center). The quantity represented is the absolute value of the neuron of the first hidden layer which is found to be exactly zero within the mini-batch before application of ReLU (exact zeros are represented in white).

**What does backpropagation compute?**  A popular thinking is that the impact of $s = \text{ReLU}'(0)$ should be limited as it concerns the value at a single point. This happens to be exact in theory but for surprisingly complex reasons related to Whitney stratification (see Section 2 and references therein):

—  For a given neural network architecture, $\text{backprop}_s$ outputs a gradient almost everywhere, independently of the choice of $s \in \mathbb{R}$, see [7] and [5] for a detailed treatment of ReLU networks.

—  Under proper randomization of the initialization and the step-size of SGD, with probability 1 the value of $s$ has no impact during training with $\text{backprop}_s$ as a gradient surrogate, see [8, 6].

In particular, the set of network parameters such that $\text{backprop}_0 \neq \text{backprop}_1$ is negligible and the vast majority of SGD sequences produced by training neural networks are not impacted by changing the value of $s = \text{ReLU}'(0)$. These results should in principle close the question about the role of $\text{ReLU}'(0)$. Yet, this is not what we observe in practice with the default settings of usual libraries.

**A surprising experiment and the bifurcation zone:**  An empirical observation on MNIST triggered our investigations: consider a fully connected ReLU network, and let $(\theta_{k,0})_{k\in\mathbb{N}}$ and $(\theta_{k,1})_{k\in\mathbb{N}}$ be two training weights sequences obtained using SGD, with the same random initialization and the same mini-batch sequence but choosing respectively $s = 0$ and $s = 1$ in PyTorch [26]. As depicted in Figure 1 (left), the two sequences differ. A closer look shows that the sudden divergence is related to what we call the *bifurcation zone*, i.e., the set $S_{0,1}$ of weights such that $\text{backprop}_0 \neq \text{backprop}_1$. As mentioned previously this contradicts theory which predicts that the bifurcation zone is met with null probability during training. This contradiction is due to the precision of floating point operations and, to a lesser extent, to the size of deep learning problems. Indeed, rounding schemes used for inexact arithmetics over the reals (which set to zero all values below a certain threshold), may "thicken" negligible sets. This is precisely what happens in our experiments (Figure 1).

**The role of numerical precision:**  Contrary to numerical linear algebra libraries such as `numpy`, which operates by default under 64 bits precision, the default choice in PyTorch is 32 bits precision (as in TensorFlow or Jax). We thus modulated machine precision to evaluate the importance of the bifurcation zone in Section 3. In 32 bits, we observed that empirically the zone occupies from about 10% to 90% of the network weight space. It becomes invisible in 64 bits precision even for quite large architectures, while, in 16 bits, it systematically fills up the whole space. Although numerical precision is the primary cause of the apparition of the zone, we identify other factors such as network size, sample size. Let us mention that low precision neural network training is a topic of active research [33, 20, 11, 16], see also [28] for an overview. Our investigations are complementary as we focus on the interplay between nonsmoothness and numerical precision.

**Impact on learning:**  The next natural question is to measure the impact of the choice of $s = \text{ReLU}'(0)$ in machine learning terms. In Section 4, we conduct extensive experiments combining different architectures (fully connected, VGG, ResNet), datasets (MNIST, SVHN, CIFAR10,

ImageNet) and other learning factors (Adam optimizer, batch normalization, dropout). In 32 bits numerical precision (default in PyTorch or Tensorflow), we consistently observe an effect of choosing $s \neq 0$. We observe a significant decrease in terms of test accuracy as $|s|$ increases; this can be explained by chaotic oscillatory behaviors induced during training. In some cases gradients even explode and learning cannot be achieved. The sensitivity to this effect highly depends on the problem at hand, in particular, on the network structure and the dataset. On the other hand the choice $s = 0$ seems to be the most stable. We also observe that both batch normalization [21] and—to a lesser degree—the Adam optimizer [22] considerably mitigate this effect. All our experiments are done using PyTorch [26]; we provide the code to generate all figures presented in this manuscript.

One important message is that, even if the default choice $s = 0$ seems to be the most stable, our experiments show a counter-intuitive phenomenon that illustrates the interplay between numerical precision and nonsmoothness, and calls for caution when learning nonsmooth networks.

**Outline of the paper:** In Section 2 we recall elements of nonsmooth algorithmic differentiation which are key to understand the mathematics underlying our experiments. Most results were published in [7, 8]; we provide more detailed pointers to this literature in Appendix A.1. In Section 3 we describe investigations of the bifurcation zone and factors influencing its importance using fully connected networks on the MNIST dataset. Neural network training experiments are detailed in Section 4 with further experimental details and additional experiments reported in Appendix D.

## 2 On the mathematics of backpropagation for ReLU networks

This section recalls recent advances on the mathematical properties of backpropagation, with in particular the almost sure absence of impact of $\mathrm{ReLU}'(0)$ on the learning phase (assuming exact arithmetic over the reals). The main mathematical tools are conservative fields developed in [7]; we provide a simplified overview which is applicable to a wide class of neural networks.

### 2.1 Empirical risk minimization and backpropagation

Given a training set $\{(x_i, y_i)\}_{i=1\ldots N}$, the supervised training of a neural network $f$ consists in minimizing the empirical risk:

$$\min_{\theta \in \mathbb{R}^P} \quad J(\theta) := \frac{1}{N} \sum_{i=1}^{N} \ell(f(x_i, \theta), y_i) \tag{1}$$

where $\theta \in \mathbb{R}^P$ are the network's weight parameters and $\ell$ is a loss function. The problem can be rewritten abstractly, for each $i = 1, \ldots, N$ and $\theta \in \mathbb{R}^P$, $\ell(f(x_i, \theta), y_i) = l_i(\theta)$ where the function $l_i \colon \mathbb{R}^P \to \mathbb{R}$ is a composition of the form

$$l_i = g_{i,M} \circ g_{i,M-1} \circ \ldots \circ g_{i,1} \tag{2}$$

where for each $j = 1, \ldots, M$, the function $g_{i,j}$ is locally Lipschitz with appropriate input and output dimensions. A concrete example of what the functions $g_{i,j}$ look like is given in Appendix A.2 in the special case of fully connected ReLU networks. Furthermore, we associate with each $g_{i,j}$ a generalized Jacobian $J_{i,j}$ which is such that $J_{i,j}(w) = \mathrm{Jac}_{g_{i,j}}(w)$ whenever $g_{i,j}$ is differentiable at $w$ and Jac denotes the usual Jacobian. The value of $J_{i,j}$ at the nondifferentiability loci of $g_{i,j}$ can be arbitrary for the moment. The backpropagation algorithm is an automated implementation of the rules of differential calculus: for each $i = 1, \ldots, N$, we have

$$\mathrm{backprop}\, l_i(\theta) = J_{i,M}\left(g_{i,M-1} \circ \ldots \circ g_{i,1}(\theta)\right) \times J_{i,M-1}\left(g_{i,M-2} \circ \ldots \circ g_{i,1}(\theta)\right) \times \ldots \times J_{i,1}(\theta). \tag{3}$$

Famous autograd libraries such as PyTorch [26] or TensorFlow [1] implement dictionaries of functions $g$ with their corresponding generalized Jacobians $J$, as well as efficient numerical implementation of the quantities defined in (3).

### 2.2 ReLU networks training

Our main example is based on the function $\mathrm{ReLU} \colon \mathbb{R} \to \mathbb{R}$ defined by $\mathrm{ReLU}(x) = \max\{x, 0\}$. It is differentiable save at the origin and satisfies $\mathrm{ReLU}'(x) = 0$ for $x < 0$ and $\mathrm{ReLU}'(x) = 1$ for $x > 0$.

The value of the derivative at $x = 0$ could be arbitrary in $[0, 1]$ as we have $\partial \text{ReLU}(0) = [0, 1]$, where $\partial$ denotes the subgradient from convex analysis. Let us insist on the fact that any value within $[0, 1]$ has a variational meaning. For example PyTorch and TensorFlow use $\text{ReLU}'(0) = 0$.

Throughout the paper, and following the lines of [8], we say that a function $g : \mathbb{R}^p \to \mathbb{R}^q$ is *elementary log-exp* if it can be described by a finite compositional expression involving the basic operations $+, -, \times, /$ as well as the exponential and logarithm functions, inside their domains of definition. Examples include the logistic loss $\log(1 + e^{-x})$ on $\mathbb{R}$, the multivariate Gaussian density $(2\pi)^{-K/2} \exp(-\sum_{k=1}^{K} x_k^2/2)$ on $\mathbb{R}^K$, and the softmax function $\left(e^{x_i} / \sum_{k=1}^{K} e^{x_k}\right)_{1 \leq i \leq K}$ on $\mathbb{R}^K$. The expressions $x^3/x^2$ and $\exp(-1/x^2)$ do not fit this definition because evaluation at $x = 0$ cannot be defined by the formula. Roughly speaking a computer evaluating an elementary log-exp expression should not output any NaN error for any input.

**Definition 1** (ReLU network training). Assume that in (1), the function $\ell$ is elementary log-exp, the network $f$ has an arbitrary structure and the functions $g_{i,j}$ in (2) are either elementary log-exp or consist in applying the ReLU function to some coordinates of their input. We then call the problem in (1) a ReLU *network training problem*. Furthermore, for any $s \in \mathbb{R}$, we denote by $\text{backprop}_s$ the quantity defined in (3) when $\text{ReLU}'(0) = s$ for all ReLU functions involved in the composition (2).

**Other nonsmooth activation functions:** The ReLU operation actually allows to express many other types of nonsmoothness such as absolute values, maxima, minima, quantiles ($\text{med}$ for median) or soft-thresholding ($\text{st}$). For any $x, y, z \in \mathbb{R}$, we indeed have $|x| = \text{ReLU}(2x) - x$, $2\max(x, y) = (|x - y| + x + y)$, $\min(x, y) = -\max(-x, -y)$, $\text{med}(x, y, z) = \min(\max(x, y), \max(x, z), \max(y, z))$, $\text{st}(x) = \text{ReLU}(x - 1) - \text{ReLU}(-x - 1)$.

Definition 1 is thus much more general than it may seem since it allows, for example, to express convolutional neural networks with max-pooling layers such as VGG or ResNet architectures which correspond to the models considered in the experimental section (although we do not re-program pooling using ReLU). The following theorem is due to [7], with an alternative version in [8].

**Theorem 1** (Backprop returns a gradient a.e.). *Consider a* ReLU *network training problem* (1) *as in Definition 1 and $T \geq 1$. Define $S \subset \mathbb{R}^P$ as the complement of the set*

$$\left\{\theta \in \mathbb{R}^P, l_i \text{ differentiable at } \theta, \text{backprop}_s l_i(\theta) = \nabla l_i(\theta), \quad \forall i \in \{1, \ldots, N\}, s \in [-T, T]\right\}.$$

*Then $S$ is contained in a finite union of embedded differentiable manifolds of dimension at most $P - 1$ (and in particular has Lebesgue measure zero).*

Although this theorem looks natural, this is a nontrivial result about the backpropagation algorithm that led to the introduction of conservative fields [7]. It implies that all choices for $s = \text{ReLU}'(0)$ in $[0, 1] = \partial \text{ReLU}(0)$ are essentially equivalent modulo a negligible set $S$. Perhaps more surprisingly, $s$ can be chosen arbitrarily in $\mathbb{R}$ without breaking this essential property of $\text{backprop}$. The set $S$ is called the *bifurcation zone* throughout the manuscript. For ReLU network training problems, the bifurcation zone is a Lebesgue zero set and is actually contained locally in a finite union of smooth objects of dimension strictly smaller than the ambient dimension. This geometric result reveals a surprising link between backpropagation and Whitney stratifications, as described in [7, 8]. In any case the bifurcation zone is completely negligible. Note that the same result holds if we allow each different call to the ReLU function to use different values for $\text{ReLU}'(0)$.

### 2.3 ReLU network training with SGD

Let $(B_k)_{k \in \mathbb{N}}$ denote a sequence of mini-batches with sizes $|B_k| \subset \{1, \ldots, N\}$ for all $k$ and $\alpha_k > 0$ the learning rate. Given initial weights $\theta_0 \in \mathbb{R}^P$ and any parameter $s \in \mathbb{R}$, the SGD training procedure of $f$ consists in applying the recursion

$$\theta_{k+1,s} = \theta_{k,s} - \gamma \frac{\alpha_k}{|B_k|} \sum_{n \in B_k} \text{backprop}_s[\ell(f(x_n, \theta_{k,s}), y_n)], \tag{4}$$

where $\gamma > 0$ is a step-size parameter. Note that we explicitly wrote the dependency of the sequence in $s = \text{ReLU}'(0)$. According to Theorem 1 if the initialization $\theta_0$ is chosen randomly, say, uniformly in a ball, a hypercube, or with iid Gaussian entries, then with probability 1, $\theta_0$ does not fall in the bifurcation zone $S$. Intuitively, since $S$ is negligible, the odds that one of the iterates produced by the

algorithm fall on $S$ are very low. As a consequence, varying $s$ in the recursion (4) does not modify the sequence. This rationale is actually true for almost all values of $\gamma$. This provides a rigorous statement of the idea that the choice of $\mathrm{ReLU}'(0)$ "does not affect" neural network training. The following result is based on arguments developed in [8], see also [6] in a probabilistic context.

**Theorem 2** ($\mathrm{ReLU}'(0)$ does not impact training with probability 1). *Consider a* $\mathrm{ReLU}$ *network training problem as in Definition 1. Let* $(B_k)_{k \in \mathbb{N}}$ *be a sequence of mini-batches with* $|B_k| \subset \{1, \ldots, N\}$ *for all $k$ and $\alpha_k > 0$ the associated learning rate parameter. Choose $\theta_0$ uniformly at random in a hypercube and $\gamma$ uniformly in a bounded interval $I \subset \mathbb{R}_+$. Let $s \in \mathbb{R}$, set $\theta_{0,s} = \theta_0$, and consider the recursion given in* (4). *Then, with probability one, for all* $k \in \mathbb{N}$, $\theta_{k,s} = \theta_{k,0}$.

## 3  Surprising experiments on a simple feedforward network

### 3.1  $\mathrm{ReLU}'(0)$ **has an impact**

Even though the $\mathrm{ReLU}$ activation function is non-differentiable at 0, autograd libraries such as PyTorch [26] or TensorFlow [1] implement its derivative with $\mathrm{ReLU}'(0) = 0$. What happens if one chooses $\mathrm{ReLU}'(0) = 1$? The popular answer to this question is that it should have no effect. Theorems 1 and 2 provide a formal justification which is far from being trivial.

**A 32 bits MNIST experiment**   We ran a simple experiment to confirm this answer. We initialized two fully connected neural networks $f_0$ and $f_1$ of size $784 \times 2000 \times 128 \times 10$ with the same weights $\theta_{0,0} = \theta_{0,1} \in \mathbb{R}^P$ which are chosen at random. Using the MNIST dataset [24], we trained $f_0$ and $f_1$ with the same sequence of mini-batches $(B_k)_{k \in \mathbb{N}}$ (minibatch size 128), using the recursion in (4) for $s = 0$ and $s = 1$ and with a fixed $\alpha_k = 1$, and $\gamma$ chosen uniformly at random in $[0.01, 0.012]$. At each iteration $k$, we computed the sum $\|\theta_{k,0} - \theta_{k,1}\|_1$ of the absolute differences between the coordinates of $\theta_{k,0}$ and $\theta_{k,1}$. As a sanity check, we actually computed $\theta_{k,0}$ a second time, denoting this by $\tilde{\theta}_{k,0}$, using a third network to control for sources of divergence in our implementation. Results are reported in Figure 1. The experiment was run using PyTorch [26] on a CPU.

**ReLU'(0) has an impact**   First we observe no difference between $\theta_{k,0}$ and $\tilde{\theta}_{k,0}$, which shows that we have controlled all possible sources of divergence in PyTorch. Second, while no differences between $\theta_{0,0}$ and $\theta_{0,1}$ is expected (Theorem 2), we observe a sudden deviation of $\|\theta_{k,0} - \theta_{k,1}\|_1$ at iteration 65 which then increases in a smooth way. The deviation is sudden as the norm is exactly zero before iteration 65 and jumps above one after. Therefore this cannot be explained by an accumulation of small rounding errors throughout iterations, as this would produce a smooth divergence starting at the first iteration. So this suggests that there is a special event at iteration 65.

The center part of Figure 1 displays the minimal absolute value of neurons of the first hidden layer evaluated on the current mini-batch, before the application of $\mathrm{ReLU}$. It turns out that at iteration 65, this minimal value is exactly 0, resulting in a drop in the center of Figure 1. This means that the divergence is actually due to an iterate of the sequence falling exactly on the bifurcation zone. According to Theorem 2, this event is so exceptional that it should never been seen.

**Practice and Theory: Numerical precision vs Genericity**   This contradiction can be solved as follows: the minimal absolute value in Figure 1 oscillates between $10^{-6}$ and $10^{-8}$ which is roughly the value of machine precision in 32 bits float arithmetic. This machine precision value is of the order $10^{-16}$ in 64 bits floating arithmetic which is orders of magnitude smaller than the typical value represented in Figure 1. And indeed, performing the same experiment in 64 bits precision, the divergence of $\|\theta_{k,0} - \theta_{k,1}\|_1$ disappears and the algorithm can actually be run for many epochs without any divergence between the two sequences. This is represented in Figure 7 in Appendix B. We also report similar results using $\mathrm{ReLU6}$ [19] in place of $\mathrm{ReLU}$ on a similar network.

### 3.2  Relative volume of the bifurcation zone and relative gradient variation

The previous experiment suggests that mini-batch SGD algorithm (4) crossed the bifurcation zone:
$$S_{01} = \{\theta \in \mathbb{R}^P : \exists i \in \{1, \ldots, N\}, \ \mathrm{backprop}_0[l_i](\theta) \neq \mathrm{backprop}_1[l_i](\theta)\} \subset S. \qquad (5)$$
This unlikely phenomenon is due to finite numerical precision arithmetic which thickens the negligible set $S_{01}$ in proportion to the machine precision threshold. This is illustrated on the right of

Figure 1, which represents the bifurcation zone at iteration 65 in a randomly generated hyperplane (uniformly among 2 dimensional hyperplanes) centered around the weight parameters which generated the bifurcation in the previous experiment (evaluation on the same mini-batch). The white area corresponds to some entries being exactly zero, i.e., below the 32 bits machine precision threshold, before application of ReLU. On the other hand, in 64 bits precision, the same representation is much smoother and does not contain exact zeros (see Figure 7 in Appendix B). This confirms that the level of floating point arithmetic precision explains the observed divergence. We now estimate the relative volume of this bifurcation zone by Monte Carlo sampling (see Appendix C for details). All experiments are performed using PyTorch [26] on GPU.

**Experimental procedure – weight randomization:** We randomly generate a set of parameters $\{\theta_j\}_{j=1}^M$, with $M = 1000$, for a fully connected network architecture $f$ composed of $L$ hidden layers. Given two consecutive layers, respectively composed of $m$ and $m'$ neurons, the weights of the corresponding affine transform are drawn independently, uniformly in $[-\alpha, \alpha]$ where $\alpha = \sqrt{6}/\sqrt{m}$. This is the default weight initialization scheme in PyTorch (Kaiming-Uniform [17]). Given this sample of parameters, iterating on the whole MNIST dataset, we approximate the proportion of $\theta_j$ for which $\mathrm{backprop}_0(l_i)(\theta_j) \neq \mathrm{backprop}_1(l_i)(\theta_j)$ for some $i$, for different networks and under different conditions (see Appendix C for details).

**Impact of the floating-point precision:** Using a fixed architecture of three hidden layers of 256 neurons each, we empirically measured the relative volume of $S_{01}$ using the above experimental procedure, varying the numerical precision. Table 1 reports the obtained estimates. As shown in Table 1, line 1, at 16 bits floating-point precision, all drawn weights $\{\theta_i\}_{i=1}^M$ fall within $S_{01}$. In sharp contrast, when using a 64 bits precision, none of the sampled weights belong to $S_{01}$. This proportion is 40% in 32 bits precision. For the rare impacted mini-batches (Table 1 line 2), the relative change in norm is above a factor 20, higher in 16 bits precision (Table 1 line 3). These results confirm that the floating-point arithmetic precision is key in explaining the impact of $\mathrm{ReLU}'(0)$ during backpropagation, impacting both frequency and magnitude of the differences.

| Floating-point precision | 16 bits | 32 bits | 64 bits |
|---|---|---|---|
| Proportion of $\{\theta_i\}_{i=1}^M$ in $S$ (CI $\pm 5\%$) | 100% | 40% | 0% |
| Proportion of impacted mini-batches (CI $\pm 13\%$) | 0.05% | 0.0002% | 0% |
| Relative $L^2$ difference for impacted mini-batches (1st quartile, median, 3rd quartile) | $(98, 117, 137)$ | $(19, 25, 47)$ | $(0, 0, 0)$ |

Table 1: Impact of $S$ according to the floating-point precision on a fully connected neural network ($784 \times 256 \times 256 \times 256 \times 10$) on MNIST. First line: proportion of drawn weights $\theta_i$ such that at least one mini-batch results in difference between $\mathrm{backprop}_0$ and $\mathrm{backprop}_1$. CI stands for *Confidence Interval* with $5\%$ risk (Hoeffding CI, see Appendix C). Second line: overall proportion of mini-batch-weight vector pairs causing a difference between $\mathrm{backprop}_0$ and $\mathrm{backprop}_1$ (McDiarmid CI, see Appendix C). Third line: distribution of $\|\mathrm{backprop}_0 - \mathrm{backprop}_1\|_2 / \|\mathrm{backprop}_0\|_2$ for the affected mini-batch-weight vector pairs.

**Impact of sample and mini-batch size:** Given a training set $\{(x_n, y_n)\}_{n=1...N}$ and a random variable $\theta \in \mathbb{R}^P$ with distribution $\mathbb{P}_\theta$, we estimate the probability that $\theta \in S_{01} \subset S$. Intuitively, this probability should increase with sample size $N$. We perform this estimation for a fixed architecture of 4 hidden layers composed of 256 neurons each while varying the sample size. Results are reported in Figure 2. For both the 16 and the 32 bits floating-point precisions, our estimation indeed increases with the sample size while we do not find any sampled weights in $S_{01}$ in 64 bits precision. We also found that the influence of mini-batch size is not significant.

**Impact of network size:** To evaluate the impact of the network size, we carried out a similar Monte Carlo experiment, varying the depth and width of the network. Firstly, we fixed the number of hidden layers to 3. Following the same experimental procedure, we empirically estimated the relative volume of $S_{01}$, varying the width of the hidden layers. The results, reported in Figure 2, show that increasing the number of neurons by layer increases the probability to find a random $\theta \sim \mathbb{P}_\theta$ in $S_{01}$ for both 16 and 32 floating-point precision. In 64 bits, even with the largest width tested (1024 neurons), no sampled weight parameter is found in $S_{01}$. Similarly we repeated the experiment varying the network depth and fixing, this time, the width of the layers to 256 neurons. Anew, the

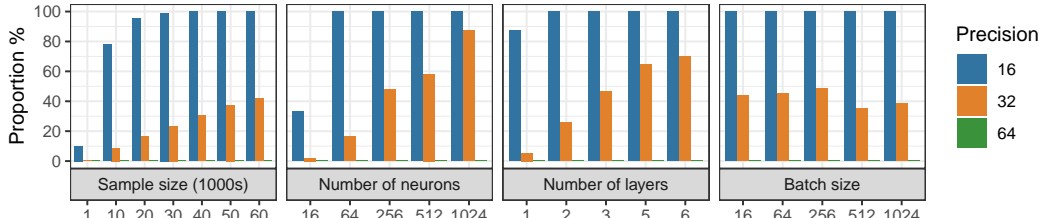

Figure 2: Influence of different size parameters on the proportion of drawn weight vector $\theta_i$ such that at least one mini-batch results in difference between $\text{backprop}_0$ and $\text{backprop}_1$ on MNIST dataset. Confidence interval at risk level $5\%$ results in a variation of $\pm 5\%$ for all represented proportions. First: 4 hidden layers with 256 neurons and batch size 256, varying the size of the training set. Second: 3 hidden layer network with mini-batch size 256, varying the number of neuron per layer. Third: 256 neurons per layer and mini-batch size 256, varying the number of layers. Fourth: 3 hidden layers with 256 neurons per layer, varying mini-batch size.

results, reported in Figure 2, show that increasing the network depth increases the probability that $\theta \sim \mathbb{P}_\theta$ belongs to $S_{01}$ for both 16 and 32 bits floating-point precision while this probability is zero in 64 bits precision. This shows that the size of the network, both in terms of number of layers and size of layers is positively related to the effect of the choice of $s$ in backpropagation. On the other hand, the fact that neither the network depth, width, or the number of samples impact the 64 bits case suggests that, within our framework, numerical precision is the primary factor of deviation.

## 4 Consequences for learning

### 4.1 Benchmarks and implementation

**Datasets and networks** We further investigate the effect of the phenomenon described in Section 3 in terms of learning using the CIFAR10 dataset [23] and the VGG11 architecture [31]. To confirm our findings in alternative settings, we also use the MNIST [24], SVHN [25] and ImageNet [12] datasets, fully connected networks (3 hidden layers of size 2048), and the ResNet18 and ResNet50 architectures [18]. Additional details on the different architectures and datasets are found in Appendix D.1. By default, unless stated otherwise, we use the SGD optimizer. We also investigated the effect of batch normalization [21], dropout [32], the Adam optimizer [22] as well as numerical precision. All the experiments in this section are run in PyTorch [26] on GPU. For each training experiment presented in this section (except ImageNet experiments), we use the `optuna` library [3] to tune learning rates for each experimental condition; see also Appendix D.2.

### 4.2 Effect on training and test errors

We first consider training a VGG11 architecture on CIFAR10 using the SGD optimizer. For different values of $\text{ReLU}'(0)$, we train this network ten times with random initializations under 32 bits arithmetic precision. The results are depicted in Figure 3. Without batch normalization, varying the value of $\text{ReLU}'(0)$ beyond a magnitude of $0.1$ has a detrimental effect on the test accuracy, resulting in a concave shaped curve with maximum around $\text{ReLU}'(0) = 0$. On the other hand, the decrease of the training loss with the number of epochs suggests that choosing $\text{ReLU}'(0) \neq 0$ induces jiggling behaviors with possible sudden jumps during training. Note that the choice $\text{ReLU}'(0) = 0$ leads to a smooth decrease and that the magnitude of the jumps for other values is related to the magnitude of the chosen value. This is consistent with the interpretation that changing the value of $\text{ReLU}'(0)$ has an excitatory effect on training. We observed qualitatively similar behaviors for a fully connected network on MNIST and a ResNet18 on CIFAR10 (Appendix D). Sensitivity to the magnitude of $\text{ReLU}'(0)$ depends on the network architecture: our fully connected network on MNIST is less sensitive to this value than VGG11 and ResNet18. The latter shows a very high sensitivity since for values above $0.2$, training becomes very unstable and almost impossible.

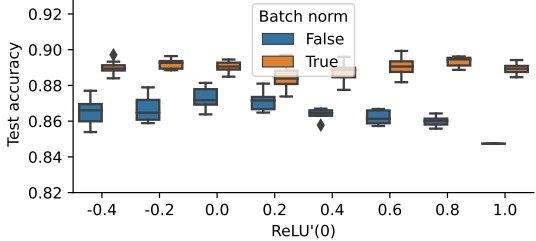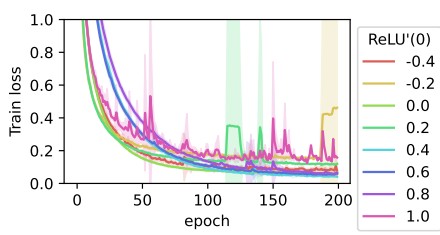

Figure 3: Training a VGG11 network on CIFAR10 with SGD. Left: test accuracy with and without batch normalization. Right: training loss without batchnorm. For each experiment we performed 10 random initializations, depicted by the boxplots on the left and the filled contours on the right (standard deviation).

These experiments complement the preliminary results obtained in Section 3. In particular choosing different values for $\mathrm{ReLU}'(0)$ has an effect, it induces a chaotic behavior during training which affects test accuracy. The default choice $\mathrm{ReLU}'(0) = 0$ seems to provide the best performances.

To conclude, we conducted four training experiments for ResNet50 on the ImageNet dataset [12] using the SGD optimizer. These were conducted with fixed learning rate, contrary to results reported above. We observe that switching from $\mathrm{ReLU}'(0) = 0$ to 1 results in a massive drop from around 75% to 63% or 55% for two runs.

### 4.3 Mitigating factors: numerical precision, batch-normalization and Adam

We analyze below the combined effects of the variations of $\mathrm{ReLU}'(0)$ with numerical precision or classical reconditioning methods: Adam, batch-normalization and dropout.

**Batch-normalization:** As represented in Figure 3, batch normalization [21] not only allows to attain higher test accuracy, but it also completely filters out the effect of the choice of the value of $\mathrm{ReLU}'(0)$, resulting in a flat shaped curve for test accuracy. This is consistent with what we observed on the training loss (Figure 12 in Appendix D.3) for which different choices of $\mathrm{ReLU}'(0)$ lead to indistinguishable training loss curves. This experiment suggests that batch normalization has a significant impact in reducing the effect of the choice of $\mathrm{ReLU}'(0)$. This observation was confirmed with a very similar behavior on the MNIST dataset with a fully connected network (Figure 9 in Appendix D.2). We could observe a similar effect on CIFAR 10 using a ResNet18 architecture (see Appendix D.5), however in this case the value of $\mathrm{ReLU}'(0)$ still has a significative impact on test error, the ResNet18 architecture being much more sensitive.

**Using the Adam optimizer:** The Adam optimizer [22] is among the most popular algorithms for neural network training; it combines adaptive step-size strategies with momentum. Adaptive step-size acts as diagonal preconditioners for gradient steps [22, 13] and therefore can be seen as having an effect on the loss landscape of neural network training problems. We trained a VGG11 network on both CIFAR 10 and SVHN using the Adam optimizer. The results are presented in Figure 4. We observe a qualitatively similar behavior as in the experiments of Section 4.2 but a much lower sensitivity to the magnitude of $\mathrm{ReLU}'(0)$. In other words, the use of the Adam optimizer mitigates the effect of this choice, both in terms of test errors and by buffering the magnitude of the sometimes chaotic effect induced on training loss optimization (Figure 13 in Appendix D.3).

**Increasing numerical precision:** As shown in Appendix D.3, using 64 bits floating precision on VGG11 with CIFAR10 cancels out the effect of $\mathrm{ReLU}'(0) = 1$, in coherence with Section 3. More specifically $\mathrm{ReLU}'(0) = 1$ in 64 bits precision obtains similar performances as $\mathrm{ReLU}'(0) = 0$ in 32 bits precision. Furthermore, the numerical precision has barely any effect when $\mathrm{ReLU}'(0) = 0$. Finally, we remark that in 16 bits with $\mathrm{ReLU}'(0) = 1$, training is extremely unstable so that we were not able to train the network in this setting.

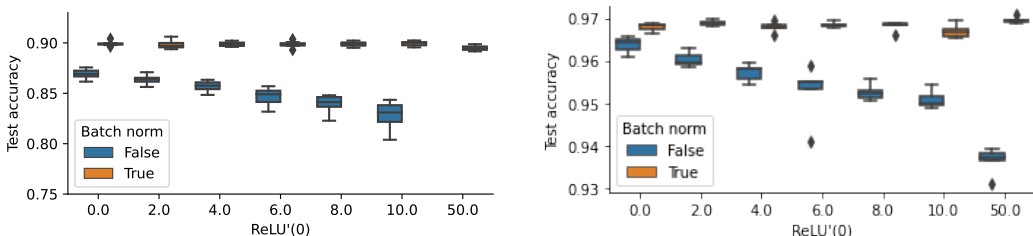

Figure 4: Similar to Fig.3, with VGG11 and Adam optimizer, on CIFAR 10 (left) and SVHN (right).

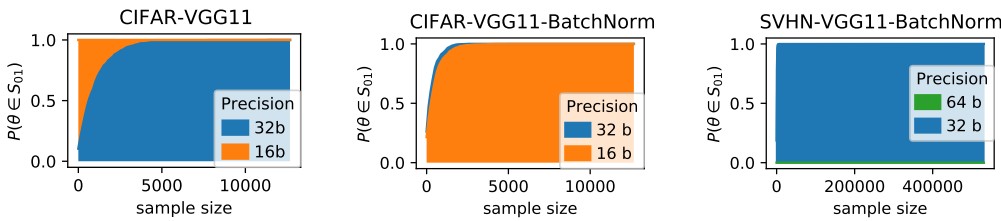

Figure 5: Monte Carlo estimation of relative volume on CIFAR10 and SVHN with VGG11 network

**Combination with dropout:** Dropout [32] is another algorithmic process to regularize deep neural networks. We investigated its effect when combined with varying choices of $\mathrm{ReLU}'(0)$. We used a fully connected neural network trained on the MNIST dataset with different values of dropout probability. The results are reported in Figure 11 in Appendix D.2. We did not observe any joint effect of dropout probability and magnitude of $\mathrm{ReLU}'(0)$ in this context.

### 4.4 Back to the bifurcation zone: more neural nets and the effects of batch-norm

The training experiments are complemented by a similar Monte Carlo estimation of the relative volume of the bifurcation zone as performed in Section 3 (same experimental setting). To avoid random outputs we force the GPU to compute convolutions deterministically. Examples of results are given in Figure 5. Additional results on fully connected network with MNIST and ResNet18 on CIFAR10 are shown in Section D. We consistently observe a high probability of finding an example on the bifurcation zone for large sample sizes. Several comments are in order.

**Numerical precision:** Numerical precision is the major factor in the thickening of the bifurcation zone. In comparison to 32 bits experiments, 16 bits precision dramatically increases its relative importance. We also considered 64 bits precision on SVHN, a rather large dataset. Due to the computational cost, we only drew 40 random weights and observed no bifurcation on any of the terms of the loss whatsoever. This is consistent with the experiments conducted in Section 3 and suggests that, within our framework, 64 bit precision is the main mitigating factor for our observations.

**Batch normalization:** In all our relative volume estimation experiments, we observe that batch normalization has a very significant effect on the proportion of examples found in the bifurcation zone. In 32 bits precision, the relative size of this zone increases with the addition of batch normalization, similar observations were made in all experiments presented in Appendix D. This is a counter-intuitive behavior as we have observed that batch normalization increases test accuracy and mitigates the effect of $\mathrm{ReLU}'(0)$. Similarly in 16 bits precision, the addition of batch normalization seems to actually decrease the size of the bifurcation zone. Batch normalization does not result in the same qualitative effect depending on arithmetic precision. These observations open many questions which will be the topic of future research.

### 4.5 Total number of $\mathrm{ReLU}$ calls during training

We consider the MNIST dataset with a fully connected $\mathrm{ReLU}$ network, varying the number of layers in $1, \ldots, 6$ and neuron per layers in $16, 64, 256, 512$. For 16 and 32 bits precisions, we perform 100

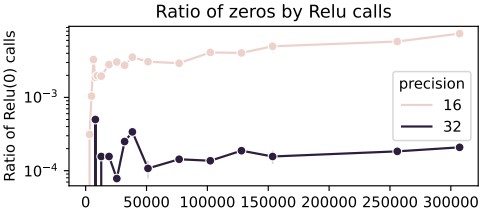

Figure 6: Proportion of $\mathrm{ReLU}(0)$ calls by total number of $\mathrm{ReLU}$ calls after 100 epochs for $\mathrm{ReLU}$ networks on MNIST with number of layers in $1, \dots, 6$ and neuron per layers in $16, 64, 256, 512$.

epochs and proportion of $\mathrm{ReLU}(0)$ over all $\mathrm{ReLU}$ calls during training. Figure 6 suggests that, for a given precision, the number of times the bifurcation zone is met is roughly proportional to the total number of $\mathrm{ReLU}$ calls during training, independently of the architecture used (number of layers, neurons per layers). This suggests that the total number of $\mathrm{ReLU}$ calls during training is an important factor in understanding the bifurcation phenomenon. Broader investigation of this conjecture will be a matter of future work.

## 5 Conclusions and future work

The starting point of our work was to determine if the choice of the value $s = \mathrm{ReLU}'(0)$ affects neural network training. Theory tells that this choice should have negligible effect. Performing a simple learning experiment, we discovered that this is false in the real world and the first purpose of this paper is to account for this empirical evidence. This contradiction between theory and practice is due to finite floating point arithmetic precision while idealized networks are analyzed theoretically within a model of exact arithmetic on the field of real numbers. Owing to the size of deep learning problems, rounding errors due to numerical precision occur at a relatively high frequency, and virtually all the time for large architectures and datasets under 32 bit arithmetic precision (the default choice for TensorFlow and PyTorch libraries).

Our second goal was to investigate the impact of the choice of $s = \mathrm{ReLU}'(0)$ in machine learning terms. In 32 bits precision it has an effect on test accuracy which seems to be the result of inducing a chaotic behavior in the course of empirical risk minimization. This was observed consistently in all our experiments. However we could not identify a systematic quantitative description of this effect; it highly depends on the dataset at hand, the network structure as well as other learning parameters such as the presence of batch normalization and the use of different optimizers. Our experiments illustrate this diversity. We observe an interesting robustness property of batch normalization and the Adam optimizer, as well as a high sensitivity to the network structure.

Overall, the goal of this work is to draw attention to an overlooked factor in machine learning and neural networks: nonsmoothness. The $\mathrm{ReLU}$ activation is probably the most widely used nonlinearity in this context, yet its nondifferentiability is mostly ignored. We highlight the fact that the default choice $\mathrm{ReLU}'(0) = 0$ seems to be the most robust, while different choices could potentially lead to instabilities. For a general nonsmooth nonlinearity, it is not clear *a priori* which choice would be the most robust, if any, and our investigation underlines the potential importance of this question. Our research opens new directions regarding the impact of numerical precision on neural network training, its interplay with nonsmoothness and its combined effect with other learning factors, such as network architecture, batch normalization or optimizers. The main idea is that mathematically negligible factors are not necessarily computationally negligible.

### Acknowledgments and Disclosure of Funding

The authors thank anonymous referees for constructive suggestions which greatly improved the paper. The authors acknowledge the support of the DEEL project, the AI Interdisciplinary Institute ANITI funding, through the French "Investing for the Future – PIA3" program under the Grant agreement ANR-19-PI3A-0004, Air Force Office of Scientific Research, Air Force Material Command, USAF, under grant numbers FA9550-19-1-7026, FA9550-18-1-0226, and ANR MaSDOL 19-CE23-0017-01. J. Bolte also acknowledges the support of ANR Chess, grant ANR-17-EURE-0010 and TSE-P.

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
