# Numerical influence of ReLU'(0) on backpropagation
## Supplementary Material

**David Bertoin**
IRT Saint Exupéry & ISAE-SUPAERO & ANITI
Toulouse, France
david.bertoin@irt-saintexupery.com

**Jérôme Bolte**
Toulouse School of Economics & Université Toulouse 1 Capitole & ANITI
Toulouse, France
jbolte@ut-capitole.fr

**Sébastien Gerchinovitz**
IRT Saint Exupéry & Institut de Mathématiques de Toulouse & ANITI
Toulouse, France
sebastien.gerchinovitz@irt-saintexupery.com

**Edouard Pauwels**
CNRS & IRIT, Université Paul Sabatier & ANITI
Toulouse, France
edouard.pauwels@irit.fr

This is the appendix for "Numerical influence of $\mathrm{ReLU}'(0)$ on backpropagation".

## A    Mathematical details for Section 2

In Section A.1, we provide some elements of proof for Theorems 1 and 2. In Section A.2, we explain how to check the assumptions of Definition 1 by describing the special case of fully connected ReLU networks.

### A.1    Elements of proof of Theorems 1 and 2

The proof arguments were described in [7, 8]. We simply concentrate on justifying how the results described in these works apply to Definition 1 and point the relevant results leading to Theorems 1 and 2.

It can be inferred from Definition 1 that all elements in the definition of a $\mathrm{ReLU}$ network training problem are piecewise smooth, where each piece is an elementary $log - exp$ function. We refer the reader to [30] for an introduction to piecewise smoothness and recent use of such notions in the context of algorithmic differentiation in [8]. Let us first argue that the results of [8] apply to Definition 1.

- We start with an explicit selection representation of $\mathrm{backprop}_s \mathrm{ReLU}$. Fix any $s \in \mathbb{R}$ and consider the three functions $f_1 \colon x \mapsto 0$, $f_2 \colon x \mapsto x$ and $f_3 \mapsto sx$ with the selection index $t(x) = 1$ if $x < 0$, 2 if $x > 0$ and 3 if $x = 0$. We have for all $x$

$$f_{t(x)} = \mathrm{ReLU}(x)$$

Furthermore, differentiating the active selection as in [8, Definition 4] we have

$$\hat{\nabla}^t f = \begin{cases} 0 & \text{if } x < 0 \\ 1 & \text{if } x > 0 \\ s & \text{if } x = 0 \end{cases}$$

and the right hand side is precisely the definition of $\text{backprop}_s$ ReLU. This shows that we have a selection derivative as used in [8].

- Given a ReLU network training problem as in Definition 1, we have the following property.
  - All elements in the ReLU network training problem are piecewise elementary $log - exp$. That is each piece can be identified with an elementary $log - exp$ function. Furthermore the selection process describing the choice of active function can similarly be described by elementary $log - exp$ functions with equalities and inequalities.

Therefore, we meet the definition of $log - exp$ selection function in [8] and all corresponding results apply to any ReLU network training problem as given in Definition 1. Fix $T \geq 1$, getting back to problem (1), using [8, Definition 5] and the selection derivative described above, for each $i = 1, \ldots, N$, there is a conservative field $D_i \colon \mathbb{R}^P \rightrightarrows \mathbb{R}^P$ (definition of conservativity is given in [7] and largely described in [8]) such that for any $s \in [0, T]$, and $\theta \in \mathbb{R}^P$

$$\text{backprop}_s l_i(\theta) \in D_i(\theta).$$

Using [7, Corollary 5] we have $D_i(\theta) = \{\nabla l_i(\theta)\}$ for all $\theta$ outside of a finite union of differentiable manifolds of dimension at most $P-1$. This leads to Theorem 1 for $s \in [0, T]$. Theorem 2 is deduced from the proof of [8, Theorem 7] (last paragraph of the proof) that with probability 1, for all $k \in \mathbb{N}$, for all $n = 1, \ldots, N$ and $s \in [0, T]$

$$\text{backprop}_s[\ell(f(x_n, \theta_{k,s}), y_n)] = \nabla_\theta \ell(f(x_n, \theta_{k,s}), y_n)$$

since we have $\theta_{0,s} = \theta_0$ for all $s$, the generated sequence in (4) does not depend on $s \in [0, T]$. This is Theorem 2 for $s \in [0, T]$, note that a similar probabilistic argument was developed in [6]. We may repeat the same arguments fixing $T < 0$, so that both results actually hold for all $s \in [-T, T]$.

## A.2 The special case of fully connected ReLU networks

The functions $g_{i,j}$ in the composition (2) can be described explicitly for any given neural network architecture. For the sake of clarity, we detail below the well-known case of fully connected ReLU networks for multiclass classification. We denote by $K \geq 2$ the total number of classes.

Consider any fully connected ReLU network architecture of depth $H$, with the softmax function applied on the last layer. We denote by $d_h$ the size of each layer $h = 1, \ldots, H$, and by $d_0$ the input dimension. In particular $d_H = K$ equals the number of classes. All the functions $f_\theta \colon \mathbb{R}^{d_0} \to \mathbb{R}^{d_H}$ represented by the network when varying the weight parameters $\theta \in \mathbb{R}^P$ are of the form:

$$f_\theta(x) = f(x, \theta) = \text{softmax} \circ A_H \circ \sigma \circ A_{H-1} \circ \cdots \sigma \circ A_1(x) \,,$$

where each mapping $A_h \colon \mathbb{R}^{d_{h-1}} \to \mathbb{R}^{d_h}$ is affine (i.e., of the form $A_h(z) = W_h z + b_h$), where $\sigma(u) = \big(\text{ReLU}(u_i)\big)_i$ applies the ReLU function component-wise to any vector $u$, and where $\text{softmax}(z) = \big(e^{z_i} / \sum_{k=1}^{d_H} e^{z_k}\big)_{1 \leq i \leq d_H}$ for any $z \in \mathbb{R}^{d_H}$. The weight parameters $\theta \in \mathbb{R}^P$ correspond to stacking all weight matrices $W_h$ and biases $b_h$ in a single vector (in particular, we have here $P = \sum_{h=1}^H d_h(d_{h-1} + 1)$). In the sequel, we set $P_h = \sum_{j=h}^H d_j(d_{j-1} + 1)$ and write $\theta_{h:H} \in \mathbb{R}^{P_h}$ for the vector of all parameters involved from layer $h$ to the last layer $H$. We also write $\text{concatenate}(x_1, \ldots, x_r)$ to denote the vector obtained by concatenating any $r$ vectors $x_1, \ldots, x_r$. In particular, we have $\theta_{h:H} = \text{concatenate}(W_h, b_h, \theta_{h+1:H})$.

Note that the decomposition above took $x$ as input, not $\theta$. We now explain how to construct the $g_{i,j}$ in (2). For each $i = 1, \ldots, N$, the function $\theta \in \mathbb{R}^P \mapsto f(x_i, \theta)$ can be decomposed as

$$f(x_i, \theta) = \text{softmax} \circ g_{i,2H-1} \circ \ldots \circ g_{i,2} \circ g_{i,1}(\theta) \,, \tag{6}$$

where, roughly speaking, the $g_{i,2h-1}$ apply the affine mapping $A_h$ to the output $z_{h-1} \in \mathbb{R}^{d_{h-1}}$ of layer $h - 1$ and pass forward all parameters $\theta_{h+1:H} \in \mathbb{R}^{P_{h+1}}$ to be used in the next layers, while the

$g_{i,2h}$ apply the ReLU function to the first $d_h$ coordinates. More formally, $g_{i,1} : \mathbb{R}^P \to \mathbb{R}^{d_1+P_2}$ is given by

$$g_{i,1}(\theta) = \text{concatenate}(W_1 x_i + b_1, \theta_{2:H}) \,,$$

$g_{i,2} : \mathbb{R}^{d_1+P_2} \to \mathbb{R}^{d_1+P_2}$ maps any $(z_1, \theta_{2:H}) \in \mathbb{R}^{d_1} \times \mathbb{R}^{P_2}$ to

$$g_{i,2}(z_1, \theta_{2:H}) = \text{concatenate}\big(\sigma(z_1), \theta_{2:H}\big)$$

and, for each layer $h = 2, \ldots, H$, the functions $g_{i,2h-1} : \mathbb{R}^{d_{h-1}+P_h} \to \mathbb{R}^{d_h+P_{h+1}}$ and $g_{i,2h} : \mathbb{R}^{d_h+P_{h+1}} \to \mathbb{R}^{d_h+P_{h+1}}$ are given by

$$g_{i,2h-1}(z_{h-1}, \theta_{h:H}) = \text{concatenate}(W_h z_{h-1} + b_h, \theta_{h+1:H})$$

and

$$g_{i,2h}(z_h, \theta_{h+1:H}) = \text{concatenate}\big(\sigma(z_h), \theta_{h+1:H}\big) \qquad \text{(for } h < H).$$

Consider now the cross-entropy loss function $\ell : \Delta(K) \times \{1, \ldots, K\} \to \mathbb{R}_+$ which compares any probability vector $q \in \Delta(K)$ of size $K$ (with non-zero coordinates $q_i > 0$) with any true label $y \in \{1, \ldots, K\}$, given by

$$\ell(q, y) = -\log q(y) \,.$$

Finally, using (6), the functions $l_i \colon \mathbb{R}^P \to \mathbb{R}$ appearing in (1)-(2) can be decomposed as

$$l_i(\theta) = \ell\big(f(x_i, \theta), y_i\big) = \big(q \in \Delta(K) \mapsto \ell(q, y_i)\big) \circ \text{softmax} \circ g_{i,2H-1} \circ \ldots \circ g_{i,2} \circ g_{i,1}(\theta) \,.$$

The last decomposition satisfies (2) with $M = 2H + 1$. Since $\ell$ is the cross-entropy loss function, all $M$ functions involved in this decomposition are either elementary log-exp or consist in applying ReLU to some coordinates of their input, and they are all locally Lipschitz, as required in Definition 1. This provides an explicit description of fully connected ReLU network and a similar description can be done for all architectures studied in this work.

## B  First experiment in 64 bits precision, and using a different activation

The code and results associated with all experiments presented in this work are publicly available here: `https://github.com/deel-ai/relu-prime`.

**64 bits precision.**  We reproduce the same bifurcation experiment as in Section 3 under 64 bits arithmetic precision. The results are represented in Figure 7 which is to be compared with its 32 bits counterpart in Figure 1. As mentioned in the main text, the bifurcation does not occur anymore. Indeed the magnitude of the smallest activation before application of ReLU is of the same order, but this time it is well above machine precision which is around $10^{-16}$. When depicting the same neighborhood as in Figure 1, the effect of numerical error completely disappears, the bifurcation zone being reduced to a segment in the picture, which is consistent with Theorems 1 and 2.

**ReLU6 activation.**  We conducted the same experiment with the ReLU6 activation function in place of ReLU and found similar results on a slightly larger network (754, 4000, 256). Recall that ReLU6 is equal to ReLU for $x < 6$ and equal to 6 for $x \geq 6$ and the default choice of derivatives at non differentiable points are zero. The illustration is given in Figure 8.

## C  Details on Monte Carlo sampling in Table 1

The code and results associated with all experiments presented in this work are publicly available here: `https://github.com/deel-ai/relu-prime`.

Recall that we want to estimate the relative volume of the set

$$S_{01} = \{\theta \in \mathbb{R}^P : \exists i \in \{1, \ldots, N\}, \ \text{backprop}_0[l_i](\theta) \neq \text{backprop}_1[l_i](\theta)\} \subset S.$$

by Monte Carlo sampling. We randomly generate a set of parameters $\{\theta_j\}_{j=1}^M$, with $M = 1000$, for a fully connected network architecture $f$ composed of $L$ hidden layers using Kaiming-Uniform [17] random weight generator. Given this sample of parameters, iterating on the whole MNIST dataset,

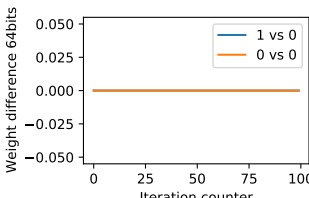 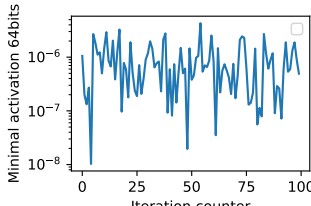 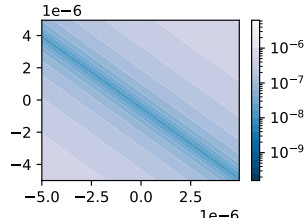

Figure 7: Same experiment as Figure 1 in 64 bits precision. Left: Difference between network parameters ($L^1$ norm), 100 iterations within an epoch. "0 vs 0" indicates $\|\theta_{k,0} - \tilde{\theta}_{k,0}\|_1$ where $\tilde{\theta}_{k,0}$ is a second run for sanity check, "0 vs 1" indicates $\|\theta_{k,0} - \theta_{k,1}\|_1$. Center: minimal absolute activation of the hidden layers within the $k$-th mini-batch, before ReLU. At iteration 65, there is no jump on the left and no drop in the center anymore. Right: illustration of the bifurcation zone at iteration $k = 65$ (same weight parameter plane as in Figure 1, but in 64 bits precision). The quantity represented is the absolute value of the neuron of the first hidden layer which was exactly zero in 32 bits (see Figure 1) before application of ReLU. Exact zeros are represented in white.

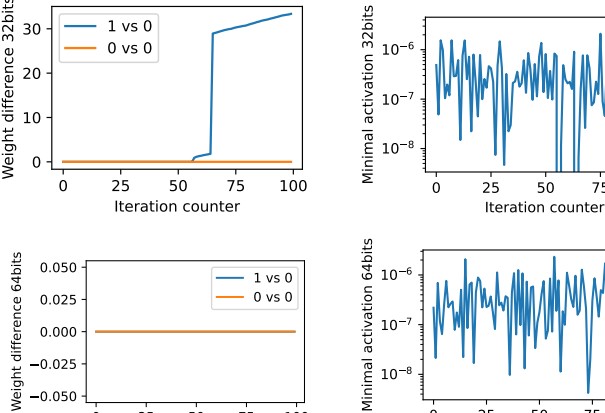

Figure 8: Same experiment as Figure 1 with ReLU6 in place of ReLU. Top: 32 bits weight difference and minimal activation before application of ReLU6. Bottom: 64 bits weight difference and minimal activation before application of ReLU6

we approximate the proportion of $\theta_j$ for which $\mathrm{backprop}_0(l_i)(\theta_j) \neq \mathrm{backprop}_1(l_i)(\theta_j)$ for some $i$, for different networks and under different conditions. More precisely, denoting by $Q$ the number of mini-batches considered in the MNIST dataset, and by $B_q \subset \{1, \ldots N\}$ the indices corresponding to the mini-batch $q$, for $q = 1, \ldots Q$, the first line of Table 1 is given by the formula

$$\frac{1}{M} \sum_{m=1}^{M} \mathbb{I}\left(\exists q \in \{1, \ldots, Q\},\ \mathrm{backprop}_0\left[\sum_{j \in B_q} l_j(\theta_m)\right] \neq \mathrm{backprop}_1\left[\sum_{j \in B_q} l_j(\theta_m)\right]\right),$$

where the function $\mathbb{I}$ takes value 1 or 0 depending on the validity of the statement in its argument. Similarly, the second line of Table 1 is given by the formula

$$\frac{1}{MQ} \sum_{m=1}^{M} \sum_{q=1}^{Q} \mathbb{I}\left(\mathrm{backprop}_0\left[\sum_{j \in B_q} l_j(\theta_m)\right] \neq \mathrm{backprop}_1\left[\sum_{j \in B_q} l_j(\theta_m)\right]\right),$$

while the last line provides statistics of the quantity

$$\frac{\left\|\mathrm{backprop}_0\left[\sum_{j \in B_q} l_j(\theta_m)\right] - \mathrm{backprop}_1\left[\sum_{j \in B_q} l_j(\theta_m)\right]\right\|}{\left\|\mathrm{backprop}_0\left[\sum_{j \in B_q} l_j(\theta_m)\right]\right\|},$$

conditioned on $q, m$ being such that $\mathrm{backprop}_0 \left[ \sum_{j \in B_q} l_j(\theta_m) \right] \neq \mathrm{backprop}_1 \left[ \sum_{j \in B_q} l_j(\theta_m) \right]$.

The error margin associated with the confidence interval on the first line of Table 1 is computed using Hoeffding's inequality at risk level 5%. It is given by the formula

$$\sqrt{\frac{\ln\left(\frac{2}{0.05}\right)}{2M}} .$$

As for the confidence interval of the second line of Table 1, we use the bounded differences inequality (a.k.a. McDiarmid's inequality) at risk level 5%. The error margin is given by the formula

$$\sqrt{\frac{1}{2}\left(\frac{1}{M} + \frac{1}{Q}\right)\ln\left(\frac{2}{0.05}\right)} .$$

# D    Complements on experiments

The code and results associated with all experiments presented in this work are publicly available here: `https://github.com/deel-ai/relu-prime`.

## D.1    Benchmark datasets and architectures

Overview of the datasets used in this work. These are image classification benchmarks, the corresponding references are respectively [24, 23, 25].

| Dataset | Dimensionality | Training set | Test set |
|---------|---------------|--------------|----------|
| MNIST | $28 \times 28$ (grayscale) | 60K | 10K |
| CIFAR10 | $32 \times 32$ (color) | 60K | 10K |
| SVHN | $32 \times 32$ (color) | 600K | 26K |
| ImageNet | $224 \times 224$ (color) | 1300K | 50K |

Overview of the neural network architectures used in this work. The corresponding references are respectively [32, 31, 18].

| Name | Type | Layers | Loss function |
|------|------|--------|---------------|
| Fully connected | fully connected | 4 | Cross-entropy |
| VGG11 | convolutional | 9 | Cross-entropy |
| ResNet18 | convolutional | 18 | Cross-entropy |
| ResNet50 | convolutional | 50 | Cross-entropy |

**Fully connected architecture:**    This architecture corresponds to the one used in [32]. We only trained this network on MNIST, the resulting architecture has an input layer of size 784, three hidden layers of size 2048 and the ouput layer is of size 10.

**VGG11 architecture:**    We used the implementation proposed in the following repository `https://github.com/kuangliu/pytorch-cifar.git` which adapts the VGG11 implementation of the module `torchvision.models` for training on CIFAR10. The only modification compared to the standard implementation is the fully connected last layers which only consist in a linear $512 \times 10$ layer. When adding batch normalization layers, it takes place after each convolutional layer.

**ResNet18 architecture:**    We use PyTorch implementation for this architecture found in the module `torchvision.models`. We only modified the size of the output layer (10 vs 1000), the size of the kernel in the first convolutional layer (3 vs 7) and replaced batch normalization layers by the identity (when we did not use batch normalization).

## D.2    Additional Experiments with MNIST and fully connected networks

We conducted the same experiments as in Section 4.2 with a fully connected 784-2048-2048-2048-10 network on MNIST. The results are represented in Figure 9 which parallels the results in Figure 3

on VGG11 with CIFAR10. We observe a similar qualitative behavior, but the fully connected architecture is less sensitive to the magnitude chosen for $\mathrm{ReLU}'(0)$. Note that in this case, learning rate tuning with optuna [3] induces a lot of spurious variability. Indeed, the same experiment with fixed learning rate results in a much smoother bell shape in Figure 10.

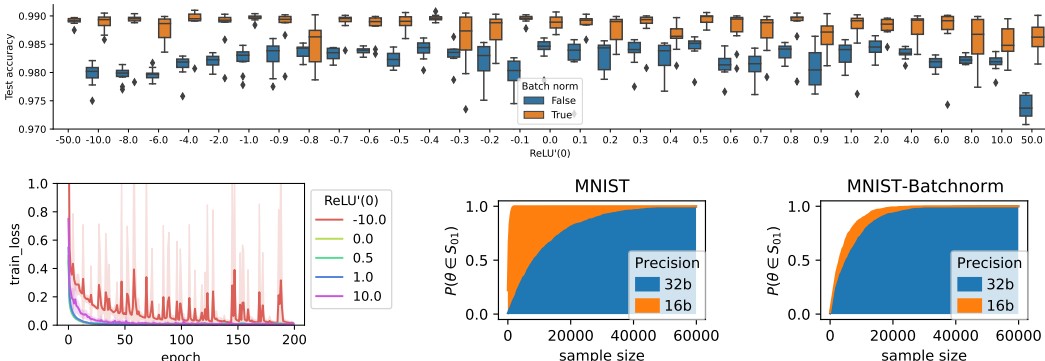

Figure 9: Top: Test error on MNIST with a fully connected 784-2048-2048-2048-10 network. The boxplots and shaded areas represent variation over ten random initializations. We recover the bell shaped curve, but the sensitivity to $\mathrm{ReLU}'(0)$ is less important. Bottom left: corresponding training loss, higher magnitude of $\mathrm{ReLU}'(0)$ induces chaotic oscillation explaining the decrease in test accuracy. Bottom center and right: relative volume estimation of the bifurcation zone without and with batch normalization. Batch normalization increases the size of the bifurcation zone with 32 bits arithmetic and decreases it under 16 bits arithmetic precision.

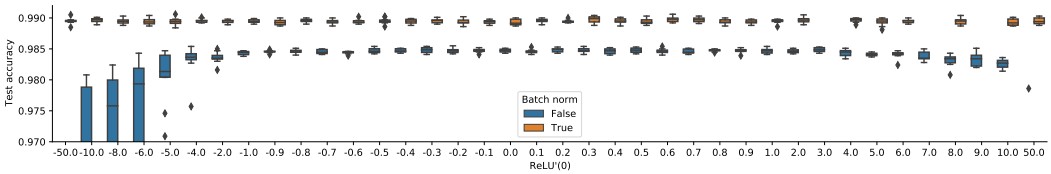

Figure 10: Same experiment as in Figure 9 without learning rate tuning.

We investigated further the effect of combining different choices of $\mathrm{ReLU}'(0)$ with dropout [32]. Dropout is another algorithmic way to regularize deep networks and it was natural to wonder if it could have a similar effect as batch normalization. Using the same network, we combined different choices of dropout probability with different choices of $\mathrm{ReLU}'(0)$. The results are represented in Figure 11 and suggests that dropout has no conjoint effect.

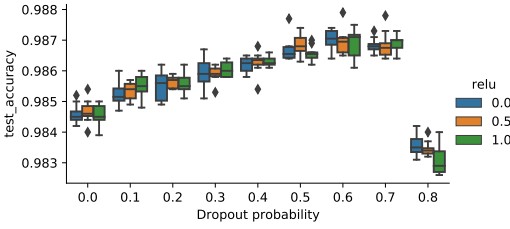

Figure 11: Experiment on combination of the choice of $\mathrm{ReLU}'(0)$ with dropout on MNIST with a fully connected 784-2048-2048-2048-10 network. The boxplots represent 10 random initializations.

## D.3 Additional experiments with VGG11

This section complements Sections 4.2 and 4.3, with additional experiments with VGG11.

**Batch normalization.**   As suggested by the experiment shown in Section 4.3, batch normalization stabilizes the choice of $\mathrm{ReLU}'(0)$, leading to higher test performances. We display in Figure 12 the decrease of training loss on CIFAR 10 and SVHN, for VGG11 with batch normalization. We see that the choice of $\mathrm{ReLU}'(0)$ has no impact and that the chaotic oscillations induced by this choice have completely disappeared.

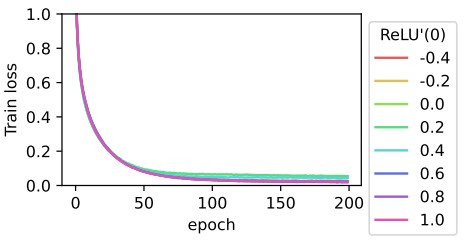 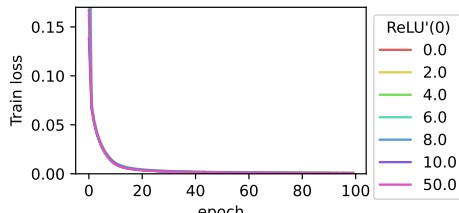

Figure 12: Training loss on CIFAR10 with VGG11 (left) and SVHN with VGG11 (right). The instability induced by the choice of $\mathrm{ReLU}'(0)$ completely disappears with batch normalization.

**Adam optimizer.**   The training curves corresponding to Figure 4 are shown in Figure 13. They suggest that the Adam optimizer features much less sensitivity than SGD to the choice of $\mathrm{ReLU}'(0)$. This is seen with a relatively efficient buffering effect on the induced oscillatory behavior on training loss decrease.

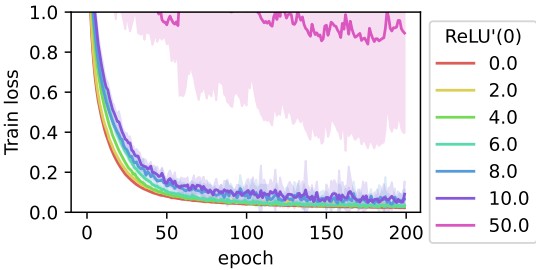 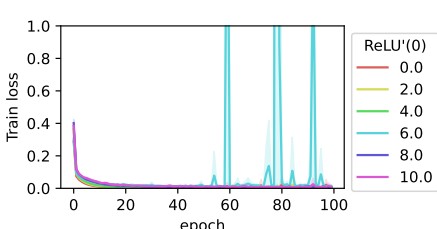

Figure 13: Training losses on CIFAR10 (left) and SVHN (right) on VGG network trained with Adam optimizer. The filled area represent standard deviation over ten random initializations.

**Numerical precision.**   For this neural network we investigated the joint effect of $\mathrm{ReLU}'(0)$ and numerical precision (16, 32 or 64 bits). The results are displayed in Figure 14. The choice $\mathrm{ReLU}'(0) = 1$ leads to such a high instability in 16 bits precision that we were not able to tune the learning rate to train the network without explosion of the weights. In 32 bits, a few experiments resulted in non-convergent training—these were removed. We observe first that for $\mathrm{ReLU}'(0) = 0$ numerical precision has barely any effect while for $\mathrm{ReLU}'(0) = 1$ it leads to an increase in test accuracy. Furthermore, we observe that $\mathrm{ReLU}'(0) = 1$ with 64 bits precision leads to the same test accuracy as $\mathrm{ReLU}'(0) = 0$ in 32 bits precision.

### D.4   Additional experiments with ResNet18

We performed the same experiments as the ones described in Section 4 using a ResNet18 architecture trained on CIFAR 10. The test error, training loss evolution with or without batch normalization are represented in Figure 15. We have similar qualitative observations as with VGG11. We note that the ResNet18 architecture is much more sensitive to the choice of $\mathrm{ReLU}'(0)$:

- Test performances degrade very fast. Actually, beyond a magnitude of 0.2, we could not manage to train the network without using batch normalization.
- Even when using batch normalization, the choice of $\mathrm{ReLU}'(0)$ seems to have an effect for relatively small variations. This is qualitatively different from what we observed with VGG11 and fully connected architectures.

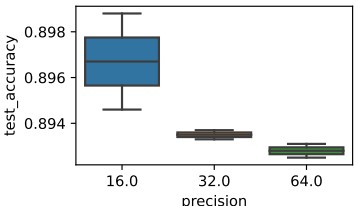 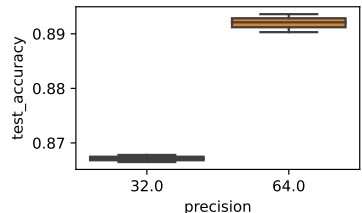

Figure 14: Test accuracy for different numerical precisions with a VGG11 network on CIFAR10. Left: $\mathrm{ReLU}'(0) = 0$. Right: $\mathrm{ReLU}'(0) = 1$.

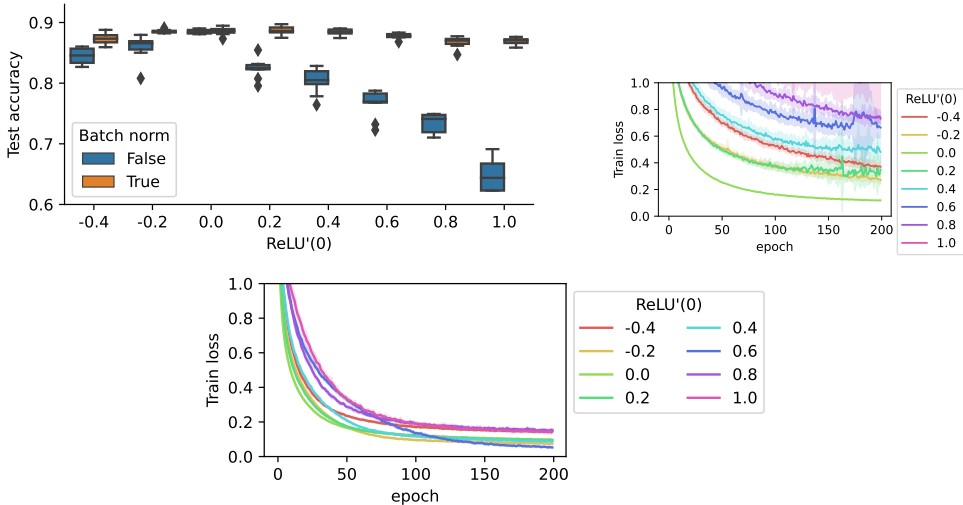

Figure 15: Training experiment on CIFAR10 with Resnet18 and the SGD optimizer. Top left: test accuracy with and without batch normalization. Top right: training loss during training without batch normalization. Bottom: training loss during training with batch normalization.

Similar Monte Carlo relative volume experiments were carried out for this network architecture; the results are presented in Figure 17. The results are qualitatively similar to what we observed for the VGG11 architecture: the bifurcation zone is met very often for 16 bits precision, and the addition of batch normalization increases this frequency in 32 bits precision. Note that we did not observe a significant variation in 16 bits precision.

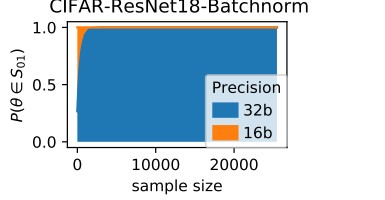 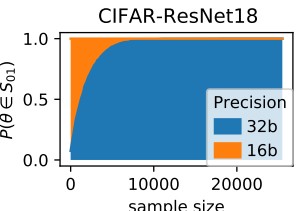

Figure 16: Relative volume Monte Carlo estimation on CIFAR10 with Resnet18 with and without batch normalization under 16 bits or 32 bits precision.

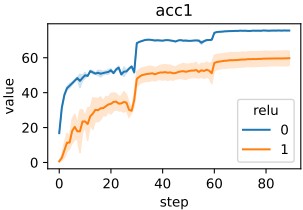

Figure 17: Test accuracy during training of a Resnet50 on ImageNet with SGD. The shaded area represents two runs. We can see a massive drop in test accuracy with $\mathrm{ReLU}'(0) = 1$.

| Dataset | Network | Optimizer | Batch size | Epochs | Time by epoch | Repetitions |
|---------|---------|-----------|------------|--------|---------------|-------------|
| CIFAR10 | VGG11 | SGD | 128 | 200 | 9 seconds | 10 times |
| CIFAR10 | VGG11 | Adam | 128 | 200 | 10 seconds | 10 times |
| CIFAR10 | ResNet18 | SGD | 128 | 200 | 13 seconds | 10 times |
| SVHN | VGG11 | Adam | 128 | 64 | 85 seconds | 10 times |
| MNIST | MLP | SGD | 128 | 200 | 2 seconds | 10 times |

Table 2: Experimental setup

### D.5 Additional experiments with ResNet50 on ImageNet

## E Complimentary information, total amount of compute and resources used

All the experiments were run on a 2080ti GPU. The code corresponding to the experiments and experiments results are available at `https://github.com/deel-ai/relu-prime` Details about each test accuracy experiments are reported on Table 2. CIFAR10 is released under MIT license, MNIST, SVHN and R are released under GNU general public license, ImageNet is released under BSD license Numpy and pytorch are released under BSD license, python is released under the python sofware fondation license.