# OpenReview forum: "Numerical influence of ReLU’(0) on backpropagation"
_NeurIPS.cc/2021/Conference — NeurIPS 2021 Poster_

### Official Review · Reviewer_7FU8 · 2021-07-15

**Rating:** 6
**Confidence:** 3

**Summary:**

This paper studies a long-neglected problem: the gradient of the non-smooth activation function in training a neural network. Specifically, the authors study the numerical influence of the ReLU derivative at zero on gradient descent. First, a theory is presented to illustrates the trivialness of ReLU'(0), then, the authors give counter-theory experiments to show that ReLU'(0) can affect training results.  In the last two sections, authors demonstrate that low-precision training, SGD optimizer, no batch norm training can magnify the impact of ReLU'(0).

**Limitations And Societal Impact:**

My major concerns are the potential lack of more experiments, I hope authors can add more results to it. Then, I'll increase my score to accept.

**Main Review:**

Overall, this is a very interesting topic that draws little attention in the community before. The authors empirically prove that the numerical choice of the non-smooth activation function's gradient can affect the training behavior. I guess this is not surprising for half-precision training since 16 bit has a much smaller representation range than single-precision values.
The paper is well-organized and well-written with a clear structure.


**Pros**:

+ The topic of this paper is quite interesting and can inspire more fellow research.

+ The experiments evaluate relevant aspects, e.g. precision bit-with, optimizer, BN, etc.

+ Paper is neat and well-written.

**Cons**:

- No large-scale experiments like ImageNet are conducted.

- I'd like to see the impact of network architectures. For example, will ResNet outperform VGG-Net in terms of this stability of the non-smooth activation function?

- Can the authors provide more examples of non-smooth activation and compare the results.


**Time Spent Reviewing:**

2

---

> ### Author Response · Authors · 2021-08-05
> **Response to 7FU8**
>
> ### Large scale experiment on ImageNet (same remark as kt6J)
> As suggested by the reviewer, we started experiments on Imagenet with a resnet50. This is extremely computationally intensive, so we can only do a few experiments. We already observe that Relu'(0) is evaluated many times on the first epoch, confirming our findings. We will wait for the final result on accuracy and include it in the paper.
>
> ### Impact of network architecture
> These results are present in the paper (Section 4, figures 3 and 11): we show that ResNet18 is more sensitive to the choice of Relu'(0) than VGG16. In particular, batch normalization completely mitigates the effect for VGG16 but not completely for ResNet18.
>
> ### Other activations  (same remark as kt6J)
> We reproduced the experiments of Section 3 using Relu6 and obtained qualitatively similar results as in the paper. This will be added in the appendix.

---

### Official Review · Reviewer_coEY · 2021-07-16

**Rating:** 7
**Confidence:** 3

**Summary:**

The ReLU function is not differentiable at 0 which has a mathematical impact on ReLU neural networks. Nevertheless they are optimized using gradient based methods. The folk wisdom is that singular points only occur with zero probability and therefore the issue is irrelevant. This paper challenges this folk wisdom and identifies that the choice of the derivative at 0 does have a tangible impact on the optimization and classification performance.

**Limitations And Societal Impact:**

yes

**Main Review:**

I think this is a good paper and such a study has been overdue.

The authors may considering the paper https://arxiv.org/pdf/1905.04992.pdf which performs a mathematical analysis of related issues.

**Time Spent Reviewing:**

2

---

> ### Author Response · Authors · 2021-08-05
> **Response to coEY**
>
> Thanks for your positive feedback. The proposed reference is very relevant and we will include a discussion about it in the main text.

---

### Official Review · Reviewer_J8rk · 2021-07-16

**Rating:** 7
**Confidence:** 3

**Summary:**

The paper demonstrates empirically that on a variety of architectures and datasets, the derivative of ReLU at 0 is evaluated a significant number of times during training in 16 and 32 bit precision. These events are frequent enough such that defining ReLU'(0) to be anything other than 0 (default value in common DNN frameworks) hurts generalization, and sometimes training as well.

The authors perform a variety of investigations around this effect, such as the influence of training set size, network width, depth, batch size, BatchNorm, Dropout.


**Limitations And Societal Impact:**

The main finding of the paper (that ReLU'(0) influences test / train loss) is potentially of limited impact, if the authors did not tune the learning rate separately for each setting, as I describe in section "Significance" above.

**Main Review:**

## Post-rebuttal Update

The authors have addressed my main concern on significance, namely that learning rate was not tuned for non-zero derivative settings. Conditioned on having the learning rate and number of training epochs tuned for each setting in section 4 (and other changes discussed in the rebuttal, e.g. the one on expected number of evaluations), I have raised the score from 5 to 7 and recommend accept, since I find it novel and potentially impactful that this setting has a strong influence not only on trainability, but also on generalization.

----
## Initial Review

Overall, I found this to be an interesting investigation with a surprising result. However, I am leaning to reject this paper due to (1) minimal qualitative / theoretical analysis or explanations of the observed phenomena (see section "Quality"), and (2) potentially flawed experimental setting to evaluate the impact of ReLU'(0) on training and test performance (see section "Significance"). I can see how the former could be improved at least a bit, and the latter can be clarified / fixed with additional experiments during rebuttal, so I am open to changing my score.

## Originality

The investigation and findings are to my knowledge original, although I am not an expert in autodiff literature.

## Quality

Mixed impressions on quality.

On one hand, I find the present experiments very thorough, well-designed, and broad, convincingly backing the claims of the paper.

On the other hand, the empirical results aren't analyzed that much, and the questions of

1. Which hyper-parameters influence the expected number of ReLU'(0) evaluations, and how?
2. Why does setting ReLU'(0) != 0 hurt performance?

remain unanswered.

For question (1), the authors could, with the data that they already have, for instance consider a conjecture that "number of ReLU'(0) evaluations is proportional to the total number of all  ReLU'(.) evaluations during training", and make the respective plot. This would explain why the number of such events grows with many hyperparameters, but not batch size.

For question (2) I don't have any immediate suggestions, apart from my comments on the learning rate in the "Significance" section.

## Clarity

Good, I found it easy to read the paper.



## Significance

Significance is at least average, since I find the fact that ReLU'(0) gets evaluated so often as to influence training / generalization quite surprising. This could make researchers more careful when reasoning about 0-probability events in floating point precision.

On the other hand,

* These results may not be of practical utility, since most frameworks already use ReLU'(0) = 0, which is empirically the best setting.
* How did you tune your learning rate, when evaluating test / training performance of networks? If the learning rate was tuned once on ReLU'(0) = 0 architectures, then it is not surprising that altering the gradients while keeping the learning rate fixed leads to a gradual decrease in performance in the magnitude of the change. In this case, I would recommend the authors to tune the learning rate separately for each value of ReLU'(0), and compare the results then. If test / train performance still varies substantially (or if that's what the authors are already doing - please clarify), then I would consider this finding quite significant, and warranting follow-up research into why this happens.


**Time Spent Reviewing:**

4

---

> ### Author Response · Authors · 2021-08-05
> **Response to J8rk**
>
> ### Conjecture on number of evaluations
> We conducted the experiment proposed by the reviewer on a fully connected network for MNIST, varying the network width and depth. The intuition of the referee seems to be correct. We observe that, for a fixed arithmetic precision (16 bits or 32 bits), the number of Relu'(0) evaluation is roughly proportional to the total number of Relu evaluations during training. Furthermore, the proportion is higher in 16 bits precision than 32 bits. These results will be included in the paper with experimental details and graphical visualization in the Appendix. We want to thank the reviewer for this very interesting suggestion which constitutes an additional element of explanation for our findings.
>
> ### Practical utility
> We found that ReLU'(0) is a hyperparameter ... whose default choice is indeed "optimal"! However, this might not be the case for other nonlinearities such as max-pooling or ReLU6  (note that the default choice of ReLU6'(6) is 1 in Tensorflow and Pytorch). We think that this opens the way to new research directions.
>
> ### Learning rate tuning
> We did not consider learning rate as a hyperparameter in the experiments; it was very roughly tuned so that training converges in all settings (with the same learning rate). As suggested by the referee, we did additional experiments to account for learning rate tuning. More specifically, using VGG16 on CIFAR10, we conducted parameter search on the learning rate (with Optuna) for both Relu'(0) = 0 and Relu'(0) = 1. For simplicity, we considered constant learning rates. We evoke below the result for one training experiment, and we will include several repeats in the paper with experimental details in the Appendix. The results confirm the trend we have observed already and show that Relu'(0) = 1 introduces excitation/instability in the dynamics, requiring a smaller learning rate (see next comment).
>     * ReLU'(0) = 1: best accuracy (84.1%), learning rate (0.0014)
>     * ReLU'(0) = 0: best accuracy (87.4%), learning rate (0.02)
> We want to thank once more the referee for this remark as this experiment confirms our findings.
>
> ### Why does setting ReLU'(0) != 0 hurt performance?
> Our intuition is that it sporadically gives "gradients" with larger norms, which makes training unstable (see training loss plots and the response to kt6J about 16 bits precision training with ReLU'(0) = 1). It is an excitatory mechanism that seems to have a detrimental effect on generalization. It is difficult to express this theoretically as this intuition-observation greatly depends on the dataset, the architecture, and the algorithm. For example, batch-norm seems to be mitigating this effect for the VGG16 network and not for ResNet18.

---

> > ### Comment · Reviewer_J8rk · 2021-08-23
> > **Intrigued by new results!**
> >
> > Thank you for your replies!
> >
> > I find your preliminary results on learning rate tuning quite exciting, so I'm raising my score to 6. If such a study were already included, or if you could produce a similar plot to Figure 3 for the case where each setting of ReLU'(0) has a carefully tuned learning rate on a validation set, I might consider raising it even further.
> >
> > To confirm,
> >
> > * was the best learning rate of `0.0014` the smallest one considered, or not? (If it's the smallest learning rate considered, it could be that generalization could be improved further by going even lower)
> >
> > * What are the training accuracies of these networks? I'm curious if the problem here is trainability or generalization.
> >
> > * Did both of these networks "flatten-out" in terms of training and validation accuracies (to exclude the case of `ReLU'(0) = 1` just needing more epochs)?

---

> > > ### Author Response · Authors · 2021-08-26
> > > **Learning rate tuning experiments**
> > >
> > > We conducted more extensive experiments about learning rate tuning, with constant learning rates on VGG16 and CIFAR 10. We proceeded as follows, for each value of relu'(0):
> > > - use Optuna to select an optimal learning rate;
> > > - perform 5 runs of training with the selected learning rate.
> > >
> > > We verified the caveats mentioned by the reviewer: the train and test accuracy did flatten out (modulo chaotic jumps for large values of relu'(0) on some runs, which we already observed) and we did not saturate the learning rate lower bound (which is 1e-5). Some runs did not converge, resulting in NAs or large loss and test accuracy around 0.1 (random prediction), most cases were for negative relu'(0) values (-0.2, -0.4) or large positive values (0.6, 0.8). This is consistent with our previous observations. Removing these runs, we obtain the following results, which confirms our preliminary experiment (accuracies are averaged over the successful runs).
> > >
> > > | ReLU'(0)| Test Accuracy | Train accuracy | best lr   |
> > > | --------|---------------| ---------------| ----------|
> > > | -0.4    | 0.8672        |  0.9775        |  0.010260 |
> > > | -0.2    | 0.8740        |  0.9481        |  0.025512 |
> > > |  0.0    | 0.8702        |  0.9796        |  0.007700 |
> > > |  0.4    | 0.8656        |  0.9856        |  0.002879 |
> > > |  0.6    | 0.8592        |  0.9755        |  0.003140 |
> > > |  0.8    | 0.8472        |  0.98          |  0.002430 |
> > > |  1.0    | 0.8327        |  0.9429        |  0.001133 |
> > >
> > > These confirm the preliminary experiment on learning rate tuning which we already mentioned.
> > > A figure will be included in the paper (we cannot show it to the reviewer due to Neurips anonymity rules).

---

> > > > ### Comment · Reviewer_J8rk · 2021-08-31
> > > > **Thank you for the update**
> > > >
> > > > Thank you for the update.
> > > >
> > > > I find this quite convincing and interesting to recommend accept, conditioned on all the changes discussed above (specifically, conjecture on # of evaluation + section 4 should be re-written with learning rate / duration of training tuned individually for every setting of `ReLU'(0)`).
> > > >
> > > > For future work, it would be very interesting to understand better why this performance degradation occurs or at least to connect it to some other heuristics (e.g. perhaps with custom slopes weights grow too large for some reason, and l2-regularization might help).
> > > >
> > > > P.S.: is the table above missing setting `0.2`?

---

> > > > > ### Author Response · Authors · 2021-09-01
> > > > > **Thanks!**
> > > > >
> > > > > ### Response to j8rk
> > > > >
> > > > > Yes 0.2 is missing, it is about to be corrected.
> > > > >
> > > > > We thank you for your constructive comments and for the positive feedback on the additional experiments! We will, of course, implement all the modifications you suggested.

---

### Official Review · Reviewer_kt6J · 2021-07-18

**Rating:** 3
**Confidence:** 4

**Summary:**

This paper analyzes the impact of the choice of ReLU'[0] on the fidelity of neural networks. Even though one would expect that the value of ReLU'[0] should not have any noticeable impact, the authors notice the divergence in learned parametric values that depend on this choice. They also notice that this effect is more pronounced at lower precision (due to rounding effects that are dominant at fp16 and fp32 vs. fp64). They also study this effect for neural network sizes, topologies (including networks with and without batch normalization) and datasets.

**Ethical Concerns:**

There are no ethical concerns here - beyond the use of Artificial Intelligence (AI) in general.

**Limitations And Societal Impact:**

No comments here.

**Main Review:**


The author does a good job explaining the theory and experiments associated with the study of the impact of ReLU'[0]. The challenge in my view is the following:
[1] The studies seemed to indicate that the default values of ReLU'[0] = 0 seem to yield the best accuracy results! Therefore, its hard to see how the outcome of this work would be in general useful to neural network (and framework) designers.
[2] The authors describe the proportion difference % between the different precisions. Can the authors comment on the final model accuracy difference for the different precisions for the different datasets. If there is no substantial difference in accuracy between the different precisions (as is indeed expected), then can the authors comment on why this result is important?
[3] In general, it'd be good to study this effect on larger models and datasets (e.g. ResNet50 or EfficientNetBx on ImageNet) - to see if this effect results in any noticeable difference in accuracy and to see if there are better optimal values for ReLU'[0]?
[4] Do the authors have results on other discontinuities besides ReLU'[0]? For e.g. one would expect ReLU[6] (used in MobileNets) to have a discontinuity at 0 and 6?


**Time Spent Reviewing:**

3

---

> ### Author Response · Authors · 2021-08-05
> **Response to kt6J:**
>
> * [1] Default choice for ReLU'(0) indeed turns out to be optimal.
> But our first finding is that this value has a genuine impact that seems unknown by all the specialists we know and contacted!
> Moreover, the default choice may not be optimal for other nonlinearities, such as max-pooling or Relu6, opening further exciting research directions. For instance, we discovered that the default choice of Relu6'(6) is 1 in TensorFlow and PyTorch (many thanks for pointing out ReLU6 to us), which is quite intriguing. [EDIT after posting the response: the previous sentence is a mistake, the default choice is 0 in both librairies]
> * [2] We actually already performed training experiments in 16 bits precision with a VGG16 network on CIFAR-10 as suggested by the reviewer. We did not report the result because training with ReLU'(0) = 1 under 16 bits precision was unsuccessful due to extreme instability. More precisely, we observed the following
>     * For ReLU'(0) = 0: The network doesn't learn with very small learning rates. Instead, learning occurs if the learning rate is above a certain threshold. In this case, we could train the network in 16-bit precision.
>     * For ReLU'(0) = 1: with small learning rates, the network doesn't learn. However, if we get above a certain threshold, the weights become extremely unstable, and some of them quickly go to infinity leading to NaN evaluations. Therefore, we could not actually train this network with ReLU'(0) = 1 with 16-bit precision and could not report accuracy results. As the reviewer suggests, we will comment on this observation in the main text with experimental results in the Appendix.
> * [3] As suggested by the reviewer, we started experiments on Imagenet with a resnet50. This is extremely computationally intensive, so we could only do a few experiments. However, we already observe that Relu'(0) is evaluated many times on the first epochs, confirming our findings. We will wait for the final result on accuracy and include it in the paper.
> * [4] We reproduced the experiments of Section 3 using Relu6 instead of Relu and obtained qualitatively similar results as in the paper. This will be added in the appendix.

---

> ### Author Response · Authors · 2021-09-01
> **Thanks and question**
>
> Comment:
> We would like to thank you again for sharing several suggestions and for the time spent reading our paper.
>
> In addition to the point-by-point response above, would you see any clarification that would be needed? Would you please not hesitate to point to any aspect you think is important to consider raising your score?

---

> ### Author Response · Authors · 2021-09-01
> **Further experiments (update)**
>
>
>
> ### Response to kt6J
>
> We would like to thank the reviewer for his constructive comments and  explain further how our latest experimental results address some concerns expressed in the review.
>
> Regarding [3], we conducted experiments on Imagenet with resnet50; the results are described in the general comment and confirm our findings.
>
> As for [2], we evaluated test accuracy of a VGG16 trained on CIFAR10 with different floating point precisions and values for ReLU'(0). The results (averaged over 3 repetitions) are as follows:
>
>
> | | 16 bits | 32 bits | 64 bits |
> | Relu'(0) = 0|0.897  |0.894|0.892|
> |Relu'(0) = 1|0.1|0.867 | 0.892|
>
> The precision does not affect test error when ReLU'(0) = 0. It has a dramatic effect for ReLU'(0) = 1 and under 16 bits precision, as we mentioned in our previous reply, training fails. On the other hand, the experiments suggest that 32 bits precision leads to a small decrease in test precision (actually one of the runs did not converge and we did not include it in the average). Finally, in 64 bits precision there is no difference between ReLU'(0)=0 and ReLU'(0)=1, which is consistent  with the rest of our findings.
>
> We thank the reviewer again for his time and we hope that we have addressed the bulk of his concerns.

---

### Author Response · Authors · 2021-08-05
**General comments**

This paper reveals that ReLU'(0) is a genuine hyperparameter of  Neural Networks in 32 bits precision (default precision of standard deep learning libraries). This seems to be unknown to many practitioners using ReLU daily (we did not actually meet someone who knew the phenomenon).
We are grateful to the referees for acknowledging this aspect. This is indeed our main contribution, together with the observation that the default choice ReLU'(0) = 0 seems to be the most advantageous. Even though this is the default value, we did not know this before our tests, so did the referees, and we did not see this in the literature.

The fact that default is the most stable choice raises the question: Was it a deliberate choice of software designers, or is it just the most natural choice? We do not know.

As explained in the Conclusion, this opens the way to new research directions with other activations and, more generally, nonsmooth AD in DL at medium/low precision.

---

> ### Author Response · Authors · 2021-08-14
> **Need for additional specific experiments?**
>
> Several of the reviewers expressed the need for conducting more numerical experiments on larger datasets or other aspects such as hyper-parameter tuning. As explained in our rebuttal, we already did some experiments in that direction and are currently waiting for more results on Imagenet with a resnet50 network.
>
> As suggested by the Area Chair, we would be happy to have your feedback (after reading our rebuttal) about which specific experiments you would like us to carry out. Also, do you see something that you deem important to complete the paper? We would be happy to work on it in the next few weeks.
>
> We would like to thank once again all the reviewers for the time spent reading our paper and for their suggestions.

---

> ### Author Response · Authors · 2021-08-26
> **Some surprising Imagenet experiments**
>
> We launched two resnet50 training on Imagenet, one with relu'(0) = 0, one with relu'(0) = 1. The gap between test accuracy for both cases is around 10 points!  In both cases, the network learns and stabilizes after 60 epochs. The test accuracy is slightly above 75% for relu'(0) = 0, which is consistent with the state of the art for this architecture. On the other hand, for relu'(0) = 1, while training is qualitatively similar (flattening after 60 epochs), we observe a massive drop in terms of the final test accuracy (around 64%). This is consistent with the rest of our observations and will be included in the main text. We repeated the experiment a second time to confirm our finding, particularly for relu'(0) = 1, and we obtained a similar result.

---

### Decision · Program_Chairs · 2021-09-27

**Decision:**

Accept (Poster)

**Comment:**

The ReLU nonlinearity is popular in deep learning. It's derivative at 0, ReLU’(0), is undefined. This paper presents the surprising observation that the chosen value of the derivative of ReLU at 0 has a substantial effect of the performance achieved in neural network training at typical numerical precision. After the initial round of reviews some concerns surfaced about how robust this observation is to the choice of learning rate and other hyperparameters. The authors addresses these concerns in their response and the reviewers now seem convinced the observed effect is real and meaningful. Most of the reviewers engaged in discussion with the authors and with the rest of the committee: They reached a consensus recommendation to accept the paper.

Reviewer kt6J is the sole remaining negative reviewer: Unfortunately they did not engage in discussion with the committee or the authors. Their review did not point out any serious problems that would invalidate the contribution made by the paper. I do not believe the review of kt6J provides enough reason to not accept the paper, given that the paper presents a clearly surprising and interesting result that is appreciated by the other reviewers.

Authors, please integrate all new results and discussion from the author response phase into the camera ready version of the paper.